# A cytofluorimetric analysis of a *Saccharomyces cerevisiae* population cultured in a fed-batch bioreactor

**Emanuela Palomba**[1©]**, Valentina Tirelli**[2©]**, Elisabetta de Alteriis**[3]**, Palma Parascandola**[4]**, Carmine Landi**[4]**, Stefano Mazzoleni**[5]**, Massimo Sanchez**[2]*

**1** Department of Research Infrastructures for marine biological resources (RIMAR), Stazione Zoologica "Anton Dohrn", Villa Comunale, Napoli, Italy, **2** Istituto Superiore di Sanità (ISS) Core Facilities, Rome, Italy, **3** Department of Biology, University of Naples "Federico II", Naples, Italy, **4** Department of Industrial Engineering, University of Salerno, Salerno, Italy, **5** Department of Agricultural Sciences, University of Naples "Federico II", Naples, Italy

© These authors contributed equally to this work.
* massimo.sanchez@iss.it

**Data Availability Statement:** The data are available on the public repository Flow Repository, the URL is: https://flowrepository.org/id/RvFrh23lz1PIoA8cb0yfMOqshSYSsKaoQnAwXegVmOjxo9rP7W6UzfchDWZgI0fl.

## Abstract

The yeast *Saccharomyces cerevisiae* is a reference model system and one of the widely used microorganisms in many biotechnological processes. In industrial yeast applications, combined strategies aim to maximize biomass/product yield, with the fed-batch culture being one of the most frequently used. Flow cytometry (FCM) is widely applied in biotechnological processes and represents a key methodology to monitor cell population dynamics. We propose here an application of FCM in the analysis of yeast cell cycle along the time course of a typical S. *cerevisiae* fed-batch culture. We used two different dyes, SYTOX Green and SYBR Green, with the aim to better define each stage of cell cycle during S. *cerevisiae* fed-batch culture. The results provide novel insights in the use of FCM cell cycle analysis for the real-time monitoring of *S. cerevisiae* bioprocesses.

## Introduction

The yeast *Saccharomyces cerevisiae* is widely used in many industrial processes, including those related to its fermentation capacity. It is used in the food industry (brewing, winemaking, baking and food additives), in the production of biofuel and medically relevant biomolecules for therapeutic applications [1,2].

Due to the biotechnological importance of S. *cerevisiae*, yeast cultivation strategies have been improved to optimize the maximum achievable cell density in bioreactors. In particular, to increase the biomass yield, the cultural strategy of the "extended batch" or "fed-batch" culture [3,4] has been developed to prolong the classic batch culture by a continuous or intermittent supply of fresh medium to the vessel so to achieve a high cell density [5]. This process has been traditionally used to produce baker's yeast [6].

**Funding:** Emanuela Palomba is supported by a PhD fellowship founded by Stazione Zoologica Anton Dohrn and by the NOSELF s.r.l (https://www.noself.it/) The funders had no role in study design, data collection and analysis, decision to publish, or preparation of the manuscript.

**Competing interests:** The author Emanuela Palomba is supported by a PhD fellowship founded by Stazione Zoologica Anton Dohrn and by the NOSELF s.r.l (https://www.noself.it/). The funders had no role in study design, data collection and analysis, decision to publish, or preparation of the manuscript. This does not alter our adherence to PLOS ONE policies on sharing data and materials.

Further, different mathematical models have been developed and implemented to describe *S. cerevisiae* growth in different cultural conditions, to infer on and to predict yeast performance [7–10].

Developed mainly for medical and clinical purposes, flow cytometry (FCM) is a powerful technology that is finding application in agriculture and food science, including pro-biotic research and genetically modified organism development [11].

Moreover, it has been outlined how FCM technology can support other fields such as cytogenomics [12], proteomics [13], and marine cell biology [14,15].

FCM has been successfully applied in food microbiology for the assessment of safety during all steps of the food production chain, and widely used for the analysis of alcoholic beverages and dairy products [11,16–18]. Indeed, FCM analytical approaches allow high throughput detection, quantification, monitoring and, where necessary, the separation (*i.e.* cell sorting) of physiologically diverse microbial subpopulations in liquid food samples [19].

Given the positive outcome of these applications, different analysis systems have recently become available on the market to control the entire productive process or directly the final product [18].

*S. cerevisiae* growth can be efficiently monitored by FCM through the analysis of both the cell size and different cell properties (e.g: viability, vitality, apoptotic index, free radicals production, protein and nucleic acids content). This gives the possibility to correlate cellular attributes to yeast growth performance and predict the overall outcome of the bioprocess of interest [20–22].

In particular, protein and nucleic acids content showed a correlation with the growth phase and growth rate [23,24], and with the amount of recombinant proteins produced by a yeast population growing in both continuous and fed-batch cultures [19].

It is well known that in yeast the differences in DNA content are correlated within the major phases of the cell cycle [25], so the progression of a proliferating population of yeast through the cell cycle can be monitored on the basis of the differences in DNA content and cellular size (Fig 1). In particular, FCM allows the identification of the pre-replicative phases (G0 and G1), DNA synthesis stage (S), post-replicative and mitotic (G2+M) phases. Moreover, cells with fractional DNA content typical of apoptosis can be further identified as a "sub-G1" population [25,26]. For example, the analysis of cells blocked in G0/G1 phase by using SYBR Green dye, gives information on nitrogen influence during alcoholic fermentation in S. cerevisiae [27]. By using the propidium iodide (PI), Jayakody and co-authors revealed that fermentation inhibitors impact *S. cerevisiae* population by blocking cells in G2/M phase [28]. Salma et al. [29] studied the cell cycle of S. cerevisiae in synthetic wine during viable but non-culturable state, so allowing the detection of cells which are not identified with routine laboratory methods.

Interestingly, Delobel et al., [32] used FCM to quantify the relative proportions of yeast cells in each cell cycle stage at different points of the growth curve of a population in batch culture by combining the data on cell size with the outputs obtained with different DNA binding dyes: SYTOX Green, PI, TO-PRO-3, 7- aminoactinomycin D and SYBR Green I. The authors concluded that SYTOX Green performs better than the other dyes in the identification of all the different cell cycle stages, also giving information on the percentage of cells in G0 phase, and allowing a clear discrimination between G0 and G1. Indeed, they stated that the peak commonly called "sub-G1" would not be representative of apoptotic cells but of the population fraction in G0 phase. Nevertheless, they concluded by recommending to use for yeast cell cycle analysis both SYTOX Green and SYBR Green I, under defined conditions and with appropriate reference samples [32].

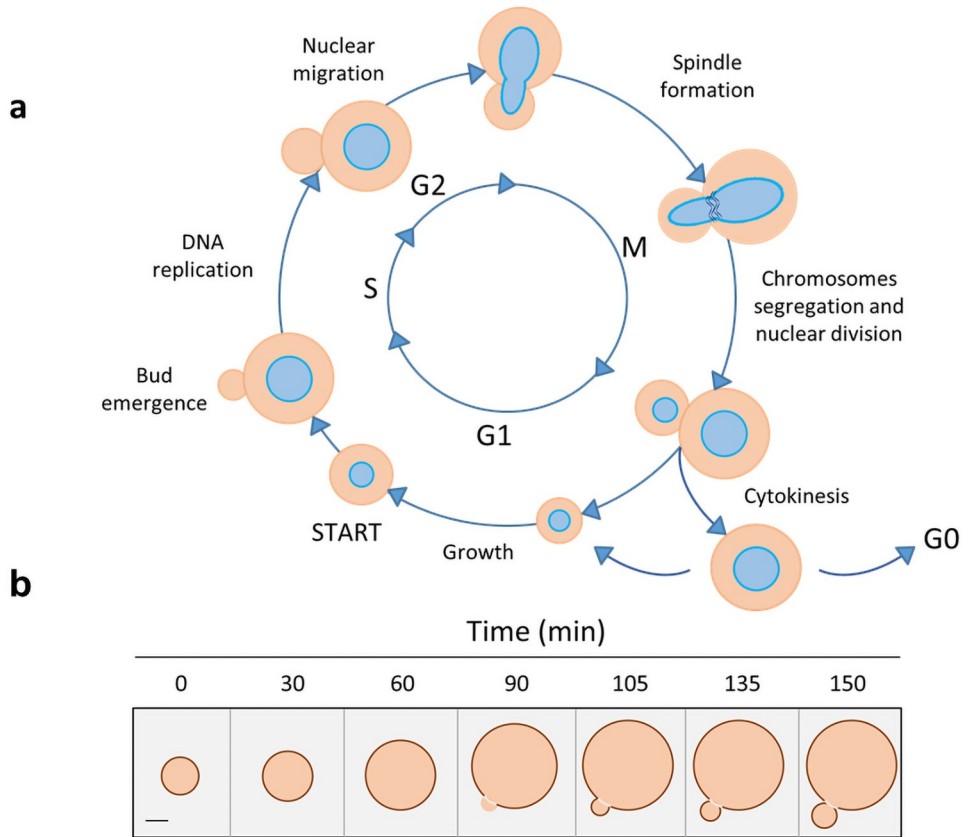

**Fig 1. Schematic view of budding yeast cell cycles: Stages (a) and size (b).** Scale bar on the left of panel B represents 2 μm. Images adapted from [30] (a) and [31] (b).

In this work, we propose a FCM analysis of yeast cell cycle along the time course of a different type of *S. cerevisiae* cultivation, the fed-batch culture, based on the use of the two recommended DNA binding dyes (SYTOX Green and SYBR Green) and cell size. By comparing the results obtained with the two dyes, we define a suitable strategy of analysis for real-time monitoring of a yeast fed-batch bioprocess.

## Materials and methods

The strain used for the experimental work was *Saccharomyces cerevisiae* CEN.PK2-1C (*MATa ura3-52 his3-D1 leu2-3, 112 trp1-289 MAL2-8c SUC2*) purchased at EUROSCARF collection (www.uni-frankfurt.de/fb15/mikro/euroscarf).

The experimental workflow is represented in Fig 2. The fed-batch culture was performed in a stirred 2 L working volume bioreactor (Bioflo 110, New Brunswick Scientific), as already described [7]. Briefly, the bioreactor filled with the medium was inoculated with an adequate aliquot of yeast pre-culture and growth was allowed to occur in batch mode. After 15 h (corresponding to time 0 of feeding phase), the feeding was started with a solution of 50% w/v glucose and salts, trace elements, glutamic acid and vitamins. The initial specific feeding rate was 0,16 h$^{-1}$, which was progressively decreased along the time course of the experiment, according to a logistically decreasing specific growth rate, as predicted by the model by Mazzoleni et al. [7].

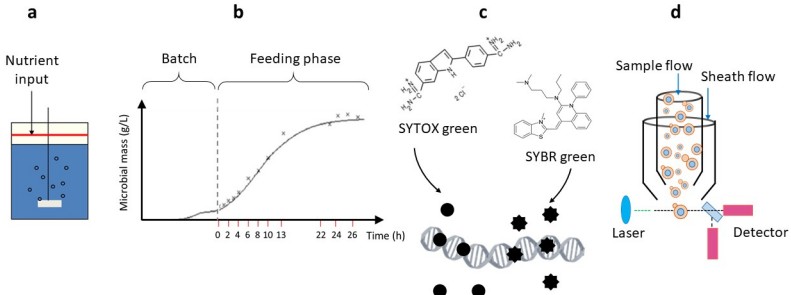

**Fig 2. Overview of the experimental workflow.** A fed-batch culture of *Saccharomyces cerevisiae* was performed in a stirred bioreactor (a) and sampled at different times during the cultivation starting from time 0 corresponding to a 15 h batch-cultivation (b). Fixed cells were stained (c) with either SYTOX Green or SYBR Green dyes for DNA detection. Finally, cells were analyzed by flow cytometer (d) as described in Materials and Methods.

Cell samples were collected at different times during the cultivation run up to 26 h of the feeding phase (see also Fig 2a) to determine cell mass (optical density at 590 nm and dry weight determination) and perform FCM analysis.

In parallel to the fed-batch culture, a batch culture was set up with the same culture medium to collect yeast cells at $0.D._{590} = 0,6$ (exponential cells) and after 7 days (starved cells), to be used as reference samples in FCM analysis.

For FCM analysis, samples were centrifuged (500 $g$, 5 min) to pellet cells and discard the culture medium. Then, cells were re-suspended and fixed in 75% ethanol, added dropwise under continuous vortexing to avoid cell agglomeration.

Fixed cell were centrifuged, treated with 1 mg ml$^{-1}$ DNase-free RNAse A (Sigma) and stained with SYTOX Green (1 μM, Invitrogen™, λex 504 nm/ λem 523 nm) or SYBR Green (1 μM, Invitrogen™, λex 497 nm/λem 518 nm). Cells were acquired by Gallios Flow cytometer, equipped with 3 lasers (405 nm, 488 nm, 633 nm, Beckman Coulter, Milan, Italy) and data were analysed with Kaluza Analysis Software v. 2.1 (Beckman Coulter).

## Results

### Identification of cell cycle stages in a fed-batch culture of S. *cerevisiae*

The fed-batch culture, which allowed yeast population to increase up to a maximal value of biomass, was sampled at different times of the feeding phase (from 0 to 26 h). From each sample, cells were isolated and stained either with SYBR or SYTOX Green dyes in order to assess the dynamic changes of DNA content during S. *cerevisiae* cell cycle (Fig 2) which together with the evaluation of cell size allowed the identification of the different cell cycle phases.

In parallel, both stains were used to identify cell cycle profiles of exponential and starved yeast cells. In particular, the exponential cells, collected from a 15 h batch culture, was regarded as reference sample (Fig 3). Here, the distribution of cell sizes (forward scatter, FSC-A) and the content of cellular DNA (green fluorescence, FL1-A) individually plotted *vs* cell count or combined in dot plots (FSC-A *vs* FL1-A) are reported for exponential (Fig 3a) and starved cells (Fig 3b), respectively.

For all the analysed stages, the percentage of cells in each cell cycle stage was similar for both dyes. The graphical results of SYBR Green and SYTOX Green staining for exponential cells were comparable: both dyes allowed a clear and precise definition of the cell cycle phases (G1, S, M and G2/M), as evidenced by the dot plots of FSC-A *vs* FL1-A and the histograms of

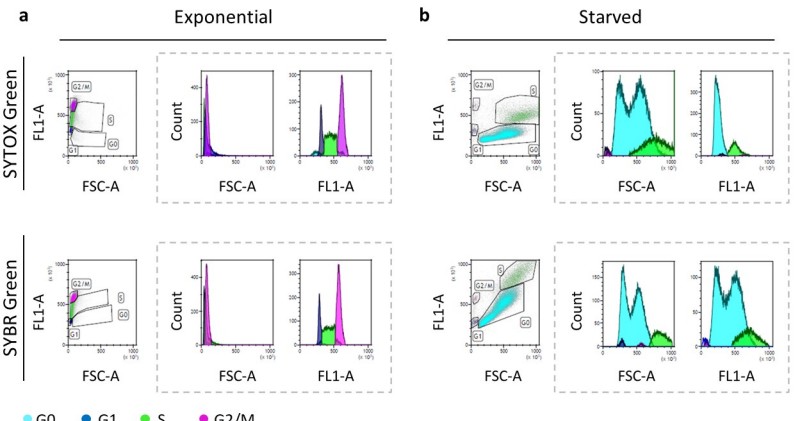

**Fig 3. Flow cytometric analysis of S. *cerevisiae* cells from a batch culture, during exponential (a) and final starvation (b) phases, stained with SYTOX Green and SYBR Green dyes.** Dashed rectangles group the mono-dimensional analysis of forward scatter signal (FSC-A) and green fluorescence (FL1-A), representing cell size and DNA content per cell, respectively, *vs* cell count. In the dot plots on the right of each panel, G0, G1, S, and G2/M cell cycle stages are identified according to both FSC-A and FL1-A.

FSC-A and FL1-A signals. The DNA content distribution of exponential cells will be used as reference for subsequent analyses.

Conversely, in the case of starved cells (Fig 3b) a more complex situation was evident. Indeed, considering the cell size (FSC-A), the staining with both SYBR Green and SYTOX Green highlighted a substantial increase and a less homogeneous distribution in both S and G0 phases. In detail, as clearly shown in the relative histograms, the FSC-A of S phase identified one population with a wider distribution of cell sizes, whereas the FSC-A of G0 phase identified two different populations with two single peaked values. Interestingly, while the cell cycle profile of samples stained with SYTOX Green was consistent with the expected distribution of DNA content, the FL1-A signal was affected by the size distribution (FCS-A signal) in samples stained with SYBR Green (Fig 3b, FL1-A histograms).

From Fig 3, it is clear that an easier and more accurate analysis of the yeast cell cycle comes from the simultaneous evaluation of DNA content and cell size (significant variable during the yeast growth). Consequently, to analyse the progression of the cell cycle over time, the mono-dimensional analysis (histogram) cannot be used alone. The bi-dimensional analysis represented by dot plots (FL1-A *vs* FSC-A), by considering also cell dimension, becomes fundamental for a clearer and more accurate interpretation of the results, thus avoiding the non-informative artefacts of mono-dimensional analysis (especially after staining with SYBR Green).

We then analysed the cell cycle phases of cells collected during the fed-batch run, represented by a yeast population grown under a continuous but progressively decreasing supply of nutrients. In Fig 4, the analysis of some representative cell samples collected at different times (0, 6, 12, 22, 26 h) during the feeding phase is shown, to make a comparison of the SYBR Green and SYTOX Green outputs. Moreover, in Fig 4 FL1-A histograms are shown in parallel to dot plots in order to confirm that the bi-dimensional analysis gives rise to an easier identification of cell cycle phases. Interestingly, the distribution of cell size in S phase gradually widens from time 0 of the feeding run, corresponding to a batch culture of a 15 h (see Material and Methods), up to 26 h, and probably was fated to widen even more reaching the distribution observed in the reference starved culture (Fig 3b). Of note, the presence of two different

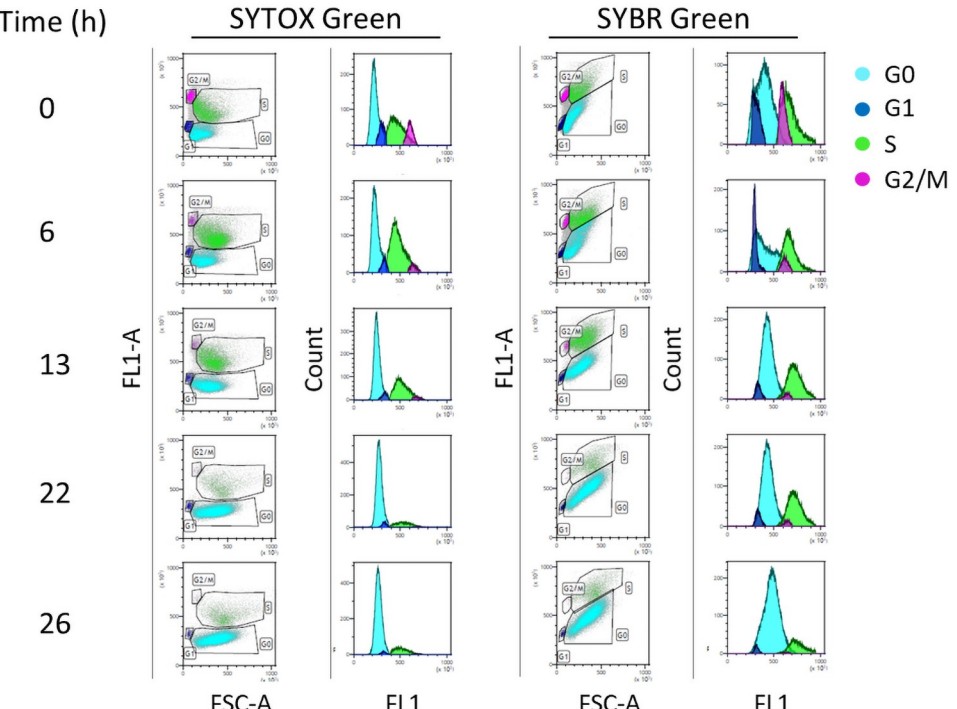

**Fig 4. Analysis of S. *cerevisiae* cell cycle during the feeding phase of the cultivation by either SYTOX Green or SYBR Green staining.** The figure shows both bi-dimensional (dot plots, FL1-A vs FSC-A) and mono-dimensional analysis (histograms of FL1-A).

populations in the G0 phase is not observed in the 0–26 h interval, probably indicating a phenomenon occurring in a more advanced culture or in starved conditions.

The green fluorescence intensity is directly proportional to the amount of DNA present in each cell, and we used the green fluorescence intensity of the exponential phase as a reference value. Considering the fluorescent signal (FL1-A on the y axis of the dot plot), in Fig 3b and in Fig 4, SYTOX Green and SYBR Green showed a different behaviour. Indeed, if we consider the characteristics of DNA content during the entire cell cycle (e.g. G2 cells have twice as much nuclear DNA as G1 cells) [25], the fluorescent signals of cells stained with SYTOX Green were more in line with those expected. Differently, when stained with SYBR Green, the fluorescence signal showed an apparent correlation with the cell size particularly in G0 and S phase where the fluorescence intensity becomes higher as the cell size increases (Figs 3b and 4).

In Fig 5 the percentage of cells in each phase of the cell cycle during the feeding phase, detected using SYTOX Green, is reported showing the overall trend over the run.

Of note, the population of cells in G0 increases with the proceeding of the feeding run while that in S phase showed an opposite trend, particularly evident from 10 h after the beginning of the run. Moreover, in the last point of the feeding run (26 h) the percentage of cells in each cell cycle phase was comparable to that of the starved phase. In detail, by comparing the values of the 26 h feeding run and those of the starved reference sample (% GO = 82,28±3,7 *vs* 81,88 ±2,94; % G1 = 3,20±0,77 and 2,43±2,05; % S = 13,61±2,45 and 15,19±4,85; % G2/M = 0,91 ±0,51 and 0,50±0,14), it is evident that the percentage of cells in each phase of the yeast

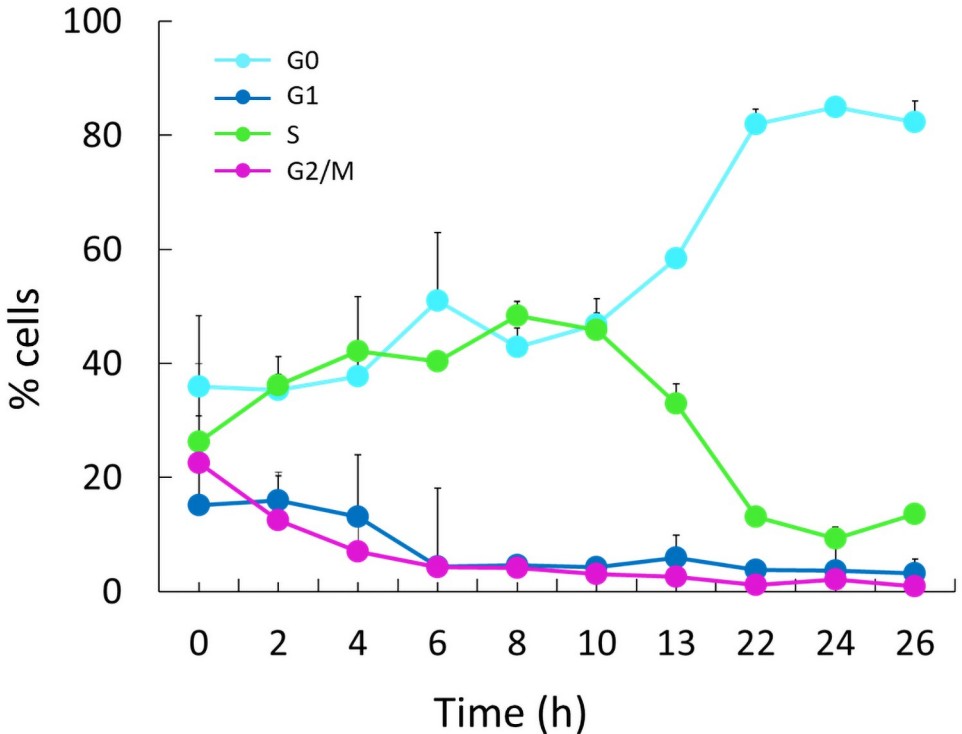

**Fig 5. Distribution of cells in different cycle stages during the feeding phase of a *S. cerevisiae* cultivation.** Only positive standard deviations are reported in the graph.

population cultured in fed-batch approached a starvation condition, in concomitance with the progressive reduction of the nutrient feeding rate along the run.

## Discussion

The yeast *Saccharomyces cerevisiae* is a reference model for biological systems widely used in many industrial applications [1,2]. The CEN.PK 2-1C strain used in the study can be considered as a reference strain. Indeed, it belongs to the CEN.PK family of isogenic laboratory strains with all possible combinations of the auxotrophic markers ura3, his3, leu2, and trp1. The CEN.PK strain family was constructed with the express aim of meeting the requirements of physiologists, geneticists, and engineers [33]. These strains display good performance in standard transformation tests and cultivation under well-defined conditions, so they are commonly used in studies related to cell growth rates and product formation, such as heterologous protein production.

In the context of industrial processes, where a critical point is the real-time monitoring of the bioprocess, FCM has been applied to control the microbial performance in bioreactors [18,34].

Recently, FCM has been used as a sensitive and reliable tool for the real-time monitoring of the relative proportion of cells for each cell cycle stage in different times of an S. *cerevisiae* batch culture [32]. Since this study recommended SYTOX and SYBR Green as most suitable DNA-binding dyes [32], we used both and the cell size parameter to determine the different phases of the cell cycle of a S. *cerevisiae* population growing in a fed-batch bioreactor and compared results to define the best method of analysis.

The bi-dimensional analysis represented by dot plots of FSC-A and FL1-A (cell size and green fluorescence, respectively) and also suggested by Zhang et al. [35] allows the rapid evaluation of two parameters the changing of which characterizes the cell cycle of budding yeast and avoids the confusing artefacts of the mono-dimensional analysis.

Our results highlighted two main features. The first one is related to the fluorescent signal. Although SYTOX Green and SYBR Green outputs are comparable in the exponential phase (Fig 3a), the SYTOX Green performs better than SYBR Green. In fact, as shown in Figs 3b and 4, the SYTOX Green staining allows to clearly identify all phases of cell cycle in yeast starved culture as well as during the whole feeding run. SYTOX Green identifies better the differences in the DNA content between S and G2/M phase, which are expected to be higher in G2/M phase [25].

Both the two dyes, SYTOX Green and SYBR Green, bind with high affinity the DNA [36,37]. The main difference is that SYBR green penetrates also fixed intact cells, while SYTOX Green easily penetrates cells with compromised membranes [38–40]. This is not a problem since cell membranes are permeabilized by the fixative process in our experiments. Further, the staining with SYBR Green has been found to be more affected than SYTOX Green by non-specific binding of the dye to sediments and debris [41–43], so the increased fluorescent signal that we found after staining yeast cells with SYBR green can be explained by a specific interaction of the dye with residual particles present in the samples. Moreover, it is known that SYBR green binds both nuclear and mitochondrial DNA [44] and it could be also possible that it binds even preferentially to mitochondrial DNA [45]. Nevertheless, further analysis is required to completely clarify the different affinity for mitochondrial and nuclear DNA of the two dyes.

The second feature is related to the cell size, and it is common to both stains: the less homogeneous distribution of cell size in S and G0 phase (Figs 3b and 4). This can be related to the gradual increase of cellular asynchrony [46–48]. Considering the fraction represented by G0 cells, the heterogeneous size is expected according to previous findings identifying in stationary cultures sub-populations characterized by different morphologic and physiological properties, i.e smaller and larger cells [49–52].

Regarding the S phase, since during that phase most of cell growth occurs in the bud [53], we can assume that the different cellular size detected in S phase depends on the different sizes of emerging buds.

Interestingly, if we consider the dimension of cells in the G1 phases as a standard for a cell after cytokinesis, from our results, we can assume that in the S phase two phenomena coexist (Figs 3b and 4). Firstly, an overall increase in cell dimension that could be dependent from a weaker control of cellular size and secondly, the growth of bud cells can be not accompanied by a proper cellular division, as previously observed [46]. Consequently, even if ready to divide, the mother and the daughter remain physically bound and the FCM device fails to consider them as two single and separate events.

Of note, it has been demonstrated that yeast cells can enter in G0 from each cell cycle phase [54]. Hence, the fact that the distribution of dimensions in G0 phase shows a profile similar to that of cells in S phase could be an evidence that the major proportion of cells in our culture entering G0 derives from S phase.

This phenomenon could probably explain the presence of two different G0 populations in the starved culture. Cells smaller in size are failing to re-enter the cell cycle while those bigger in size can represent the population of cells which exit cell cycle during the S phase. Finally, from 10 h after the beginning of the feeding phase, for each sampled time, the increment of G0 fraction and the reduction of S fraction are quantitatively comparable (Fig 5). This reinforces the hypothesis that most of the cells in G0 phase derive from S phase.

## Conclusions

In this study, the cell cycle along the time course of a S. *cerevisiae* fed-batch culture has been evaluated on the basis of cell size and DNA content variation by using the two recommended dyes SYBR Green and SYTOX Green. Despite the comparable outputs in batch exponential phase of growth, SYTOX Green staining performed better than SYBR Green in the identification of all cell cycle phases of a starved culture, as well as during the whole feeding phase of a S. *cerevisiae* fed-batch culture. Despite the difficulties in fully standardizing the analytical methods to obtain comparable results, the bi-dimensional representation has proven to be effective for characterizing the cell cycle of budding yeast grown in a fed-batch bioreactor and thus inferring on its physiological status. This could pave the way for the development of a suitable strategy of analysis in the perspective of a real-time monitoring of a yeast fed-batch bioprocess applicable with minimal effort to industrial processes.

## Author Contributions

**Conceptualization:** Elisabetta de Alteriis.

**Data curation:** Emanuela Palomba.

**Formal analysis:** Valentina Tirelli, Massimo Sanchez.

**Investigation:** Elisabetta de Alteriis, Carmine Landi.

**Methodology:** Emanuela Palomba, Valentina Tirelli, Palma Parascandola, Massimo Sanchez.

**Project administration:** Massimo Sanchez.

**Supervision:** Stefano Mazzoleni, Massimo Sanchez.

**Writing – original draft:** Emanuela Palomba, Valentina Tirelli.

**Writing – review & editing:** Emanuela Palomba, Valentina Tirelli, Elisabetta de Alteriis, Palma Parascandola, Carmine Landi, Stefano Mazzoleni, Massimo Sanchez.

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
