## [Decision Letter · Decision Letter 0]

19 Mar 2021

PONE-D-21-06261

A cytofluorimetric analysis of a Saccharomyces cerevisiae population cultured in a fed-batch bioreactor

PLOS ONE

Dear Dr. Sanchez,

Thank you for submitting your manuscript to PLOS ONE. After careful consideration, we feel that it has merit but does not fully meet PLOS ONE’s publication criteria as it currently stands. Therefore, we invite you to submit a revised version of the manuscript that addresses the points raised during the review process.

We look forward to receiving your revised manuscript.

Kind regards,

Alvaro Galli

Academic Editor

PLOS ONE

Journal Requirements:

'Emanuela Palomba is supported by a PhD fellowship founded by Stazione Zoologica Anton Dohrn and by the NOSELF s.r.l (https://www.noself.it/)'

We note that you received funding from a commercial source: NOSELF s.r.l

c. Please include your amended Competing Interests Statement and Funding statement within your cover letter. We will change the online submission form on your behalf.

Reviewers' comments:

Reviewer's Responses to Questions

**Comments to the Author**

1. Is the manuscript technically sound, and do the data support the conclusions?

Reviewer #1: Yes

Reviewer #3: No

2. Has the statistical analysis been performed appropriately and rigorously? 

Reviewer #1: Yes

Reviewer #3: No

3. Have the authors made all data underlying the findings in their manuscript fully available?

Reviewer #1: Yes

Reviewer #3: No

4. Is the manuscript presented in an intelligible fashion and written in standard English?

Reviewer #1: Yes

Reviewer #3: Yes

5. Review Comments to the Author

Reviewer #1: Review of “A cytofluorimetric analysis of a Saccharomyces cerevisiae population cultured in a fed-batch bioreactor”

I have reviewed this manuscript and find it to be a useful contribution to the methodology of bulk culture analysis. In reviewing the paper I downloaded the data using the link provided, and am glad to see that the authors are making that data available.

One concern I had is that the analysis was carried out on a single strain, and the results may differ significantly If this method is applied to a range of industrial S. cerevisiae strains. Still, it is made clear in the manuscript that this work is based on a single strain.

The manuscript is well written, and and appropriate for PLOS ONE.

Reviewer #3: This paper presents an experimental result on using the flow cytometry to monitor the distribution of yeast cells among different phases of yeast budding cycle over the course of a fed-batch cultivation. Results from two different DNA dyes, SYTOX green and SYBR green were compared, and the paper concluded that SYTOX green performs better than SYBR green.

It is not clear what contribution this paper brings. Using flow cytometry and DNA dyes to monitoring yeast budding cycle have been studied extensively in the field. In addition, there is lack of quantitative, statistical analysis to support the conclusions drawn in the paper. the major concerns are listed below:

1. The paper stated that “both dyes allowed a clear and precise definition of the cell cycle phases (G1, S, M, G2) as evidenced by the dot plots of FSC-A vs FL1-A” (page 5, line 133-135). However, the different phases depicted in Figure 3 (and other figures) are G0, G1, S and G2. None of the figures depicted the population of phase M.

2. All figures are blurry, and difficult to examine the detail.

3. There is no information on how the different phases were separated from each other. How were the boundaries among different phases determined? Were they determined in an ad hoc fashion? Why were there no cells identified for phase M? Were there any experimental validations to confirm that the sorted populations of different phases were truly the cells in the labelled phases?

4. During the exponential growth, one would expect cells from all phases of a budding cycle to show up, and the population distribution would be proportional to the duration of each phase within a cycle. It is not clear why phase M was not detected at all. Was it because the duration of phase M is extremely short, or the flow cytometer method cannot differentiate cells in phase M from other phases, such as those in phase G0?

5. In fig. 3 (b), the flow cytometry results using the two dyes showed different distribution of cell counts, particular for FL1-A, and the authors claimed that SYTOX worked better than SYBR, without any quantitative evidence. To claim one dye is better than the other, the authors should have a clear case with known population to validate their claims.

6. There is a lack of explanation and validation of the results presented. For example, why the majority of the cells were classified in G0 and S in the later part of the fed batch growth? What caused the bimodal distribution of the starved cell for phase G0?

6. PLOS authors have the option to publish the peer review history of their article (what does this mean?). If published, this will include your full peer review and any attached files.

Reviewer #1: No

Reviewer #2: No

Reviewer #3: No

---

## [Author Response · Author response to Decision Letter 0]

1 May 2021

We thank the editor and reviewers for their comments and according to that we modified our manuscript. 

Following the answers to each comment which are olso reported in the uploaded file "Response to reviewers".

Sincerely,

Dr. Massimo Sanchez

PONE-D-21-06261

Dear Editor,

please find attached our revised version of the manuscript, where we indicated all the revisions we included as suggested by the reviewers, as tracked changes in the document “Revised Manuscript with Track Changes”.

As you may see, we wish to add another author (Palma Parascandola) for his contribution to the initial experimental set up and during the revision process that is also indicated in tracked changes in the list of authors of the manuscript with track changes and in the new Cover letter.

In the following text, the answers to all criticisms and comments provided by reviewers are reported point by point.

Sincerely, 

Dr. Massimo Sanchez.

Reviewer #1: Review of “A cytofluorimetric analysis of a Saccharomyces cerevisiae population cultured in a fed-batch bioreactor”

I have reviewed this manuscript and find it to be a useful contribution to the methodology of bulk culture analysis. In reviewing the paper I downloaded the data using the link provided, and am glad to see that the authors are making that data available.

One concern I had is that the analysis was carried out on a single strain, and the results may differ significantly If this method is applied to a range of industrial S. cerevisiae strains. Still, it is made clear in the manuscript that this work is based on a single strain.

The manuscript is well written, and and appropriate for PLOS ONE.

Comment for the Reviewer #1

The CEN.PK 2-1C strain used in the study can be considered as a reference strain. Indeed, it belongs to the CEN.PK family of isogenic laboratory strains with all possible combinations of the auxotrophic markers ura3, his3, leu2, and trp1. The CEN.PK strain family was constructed with the express aim of meeting the requirements of physiologists, geneticists, and engineers (JP van D, Bauer J, Brambilla L, Duboc P, Francois JM, Gancedo C, et al. An interlaboratory comparison of physiological and genetic properties of four Saccharomyces cerevisiae strains. Enzyme Microb Techno,2000). These strains display good performance in standard transformation tests and cultivation under well-defined conditions, so they are commonly used in studies related to cell growth rates and product formation, such as heterologous protein production.

This information and relative reference on CEN.PK strain has been added to the revised manuscript from line 199 to 203.

Reviewer #3: This paper presents an experimental result on using the flow cytometry to monitor the distribution of yeast cells among different phases of yeast budding cycle over the course of a fed-batch cultivation. Results from two different DNA dyes, SYTOX green and SYBR green were compared, and the paper concluded that SYTOX green performs better than SYBR green.

It is not clear what contribution this paper brings. Using flow cytometry and DNA dyes to monitoring yeast budding cycle have been studied extensively in the field. In addition, there is lack of quantitative, statistical analysis to support the conclusions drawn in the paper. the major concerns are listed below:

Reviewer #3 Question 1 

The paper stated that “both dyes allowed a clear and precise definition of the cell cycle phases (G1, S, M, G2) as evidenced by the dot plots of FSC-A vs FL1-A” (page 5, line 133-135). However, the different phases depicted in Figure 3 (and other figures) are G0, G1, S and G2. None of the figures depicted the population of phase M.

Reviewer #3 Answer 1

The FCM analysis of cell cycle is based on measure DNA content by staining cells with SYBR Green and SYTOX green dye (in our experimental procedure) and reveals cells distribution among G1, S and G2/M phases. It is common to show the DNA distribution by histograms where the peak with higher fluorescent intensity is represented by G2/M phases. The DNA content in these two phases is the same and only by other analytical approach it is possible to clearly distinguish G2 from M (for example through the analysis of cyclins expression or microscopy for the identification of the cell cycle progression).

With the label G2, we actually mean both cells in G2 and M phases. Therefore, we thank the reviewer for its comment and according to that, we modified the images by substituting G2 with G2/M in figures and in the text. 

In the text, we made the following changes:

-Line 136 we changed “(G1, S, M and G2)” in “(G1, S and G2/M)”.

-Line 180 we changed “% G2” in “% G2/M”.

-Line 188 (in the figure 3 legends) we changed “In the dot plots on the right of each panel, G0, G1, S, and G2 cell cycle stages” with “In the dot plots on the right of each panel, G0, G1, S, and G2/M cell cycle stages”.

-From line 216 to 218: we changed the sentence “SYTOX Green identifies better the differences in the DNA content between S and G2 phase, which are expected to be higher in G2 phase” with “SYTOX Green identifies better the differences in the DNA content between S and G2/M phase, which are expected to be higher in G2/M phase”

Reviewer #3 Question 2

All figures are blurry, and difficult to examine the detail.

Reviewer #3 Answer 2. 

We apologize for the poor quality. The resolution of figures has been increased by using the Plos one suggested tool “PACE” to check the figures quality.

Reviewer #3 Question 3.1 

There is no information on how the different phases were separated from each other. How were the boundaries among different phases determined? Were they determined in an ad hoc fashion? 

Reviewer #3 Answer 3.1

The protocol consists in staining cells with a dye that binds DNA stoichiometrically and the fluorescent signal is proportional to the amount of DNA: once defined the G1 phase (the peak with low green fluorescence), the G2 phase peak is gated in a “position” with a double fluorescence. With Saccharomyces it is difficult to apply the algorithms (i.e. Michael H.Fox or J.V. Watson) that are normally used for the definition of the cell cycle with higher eukaryotic cells. We find linearity in the response during the exponential phase, which in fact we use as a reference to identify the different phases: G0, G1, S and G2/M. As indicated in the text, to precisely define the phases of the cell cycle at the different times of the culture, we propose a two-dimensional analysis that differentiates cell cycle phases on the basis of both cell size and DNA content (FSC vs green fluorescence, respectively).

Moreover, we changed the text as following:

-From line 85 to 86 we changed “based on the use of the two recommended DNA binding dyes (SYTOX Green and SYBR Green)” in “based on the use of the two recommended DNA binding dyes (SYTOX Green and SYBR Green) and the cell size”

- From line 127-128 we changed “in order to assess the dynamic changes of DNA content during S. cerevisiae cell cycle (Fig 2).” In “in order to assess the dynamic changes of DNA content during S. cerevisiae cell cycle (Fig 2) which together with the evaluation of cell size allowed the identification of the different cell cycle phases.”

-From line 208 to line 209 we changed “we used both to determine the different phases of the cell cycle of a S. cerevisiae population” in “we used both and the cell size parameter to determine the different phases of the cell cycle of a S. cerevisiae population”

Reviewer #3 Question 3.2 

Why were there no cells identified for phase M?

Reviewer #3 Answer 3.2 

The answer to this question is present in the Answer 1.

Reviewer #3 Question 3.3

Were there any experimental validations to confirm that the sorted populations of different phases were truly the cells in the labelled phases?

Reviewer #3 Answer 3.3

The work we propose is inspired by a work previously published by Delobel et al. in 2014 (A Simple FCM Method to Avoid Misinterpretation in Saccharomyces cerevisiae Cell Cycle Assessment between G0 and Sub-G1, 2014, Plos One), where different dye were compared in defining yeast cell cycle phases.

The aim of this work is to propose a method to follow the proceeding of a yeast fed-batch culture during time the as a potential real-time monitoring system in industrial application. The analysis with FCM here performed identified the different phases on the basis of two parameters: the DNA content and the cellular size by using as reference cell cycle profiles of exponential and starved yeast cells. To characterize cells in each phases, different analysis would be required (for example cell sorting followed by molecular analysis of cyclins expression in sorted populations) that we will consider to perform in future works to further validate this method.

Reviewer #3 Question 4. 

During the exponential growth, one would expect cells from all phases of a budding cycle to show up, and the population distribution would be proportional to the duration of each phase within a cycle. It is not clear why phase M was not detected at all. Was it because the duration of phase M is extremely short, or the flow cytometer method cannot differentiate cells in phase M from other phases, such as those in phase G0?

Reviewer #3 Answer 4.

Since the G2 and M phases are not clearly separated through flow cytometric analysis, what in the plot we indicate “G2” is actually an ensemble of cells that have the same probability to be in the growth phase (G2) as well as in one of the mitosis stages (M). To be more precise, we made some changes in figures and in the text according with this observation as specified in the answer 1.

Beside the FCM analysis, if the discrimination between cells in G2 and M stages is specifically required or requested during the analysis of a microbial population, additional methods are required. For instance, after sorting of cells in G2/M, the microscopic observation of cells or antibodies directed against markers specific of the M stages could be used to further define this point.

Reviewer #3 Question 5. 

In fig. 3 (b), the flow cytometry results using the two dyes showed different distribution of cell counts, particular for FL1-A, and the authors claimed that SYTOX worked better than SYBR, without any quantitative evidence. To claim one dye is better than the other, the authors should have a clear case with known population to validate their claims.

Reviewer #3 Answer 5.

From a quantitative point of view, the percentage of cells identified from the two dyes in each cell cycle phase is more or less similar. In order to better clarify this point, we made the following changes in the text:

-From line 134 to line 135: we added “For all the analyzed stages, the percentage of cells in each cell cycle stage was similar for both dyes. The graphical results of…”

The SYTOX worked better than SYBR green in that the SYBR green fluorescence intensity is not linearly correlated with that expected for some phases of cell cycle (G0 and S, particularly) producing a shift of signal becoming more evident during the observation feeding run up to the condition of starvation.

Reviewer #3 Question 6. 1

There is a lack of explanation and validation of the results presented. For example, why the majority of the cells were classified in G0 and S in the later part of the fed batch growth? 

Reviewer #3 Answer 6.1

The analysis of the cell cycle during the fed batch culture showed an increasing percentage of cells in G0 together with a decreasing of cells in S phase.

The condition of nutrient starvation in the batch culture (in the starved reference, fig 3) as well as the progressive decreasing feeding and the accumulation of toxic by-products in the fed-batch (fig. 4, 5) may block cells before the S phase and thus forcing them to enter in GO phase. Indeed, the entering into G0 phase has been demonstrated to occur from each cell cycle phase, as we reported from line 235 to line 240. From the analysis of cell size, we can identify clearly that the two populations are dimensionally similar. The hypothesis is that a part of cells, as the feeding run proceeds, were blocked in S phase and exit the cell cycle entering in G0 phase.

Reviewer #3 Question 6. 2

What caused the bimodal distribution of the starved cell for phase G0?

Reviewer #3 Answer 6.2

The bimodal distribution of G0 in the starved culture showed in our results highlighted the coexistence of populations with two different ranges of cell size within the G0 phase. The population with bigger dimension could derive from cells entering G0 by S phase while the smaller one could represent cells that are simply unable to start a new cell cycle for the lack of nutrient characterizing the starved phase of the batch culture.

To better clarify this point we added in the test from line 248 to line 250 the following sentences:

“This phenomenon could probably explain the presence of two different G0 populations in the starved culture. Cells smaller in size are failing to re-enter the cell cycle while those bigger in size can represent the population of cells which exit cell cycle during the S phase.”

Nevertheless, further investigations to clarify the reason of the bimodal distribution in G0 phase and to characterize the nature of cells are required and goes beyond the scope of this paper whose aim is to propose a real-time monitoring tool of the status of a yeast culture.

---

## [Decision Letter · Decision Letter 1]

27 May 2021

A cytofluorimetric analysis of a Saccharomyces cerevisiae population cultured in a fed-batch bioreactor

PONE-D-21-06261R1

Dear Dr. Sanchez,

We’re pleased to inform you that your manuscript has been judged scientifically suitable for publication and will be formally accepted for publication once it meets all outstanding technical requirements.

Kind regards,

Alvaro Galli

Academic Editor

PLOS ONE

Additional Editor Comments (optional):

Reviewers' comments:

Reviewer's Responses to Questions

**Comments to the Author**

1. If the authors have adequately addressed your comments raised in a previous round of review and you feel that this manuscript is now acceptable for publication, you may indicate that here to bypass the “Comments to the Author” section, enter your conflict of interest statement in the “Confidential to Editor” section, and submit your "Accept" recommendation.

Reviewer #2: All comments have been addressed

Reviewer #3: All comments have been addressed

2. Is the manuscript technically sound, and do the data support the conclusions?

Reviewer #2: Yes

Reviewer #3: (No Response)

3. Has the statistical analysis been performed appropriately and rigorously? 

Reviewer #2: Yes

Reviewer #3: (No Response)

4. Have the authors made all data underlying the findings in their manuscript fully available?

Reviewer #2: Yes

Reviewer #3: (No Response)

5. Is the manuscript presented in an intelligible fashion and written in standard English?

Reviewer #2: Yes

Reviewer #3: (No Response)

6. Review Comments to the Author

Reviewer #2: I am happy with the revision. The authors have addressed all the comments carefully. The ms can be accepted at its current format.

Reviewer #3: The authors response to my comments are acceptable, although it would be desirable that additional experiments be performed to validate their claim, i.e, the labled phases from the proposed approach are indeed what the labels indicate.

7. PLOS authors have the option to publish the peer review history of their article (what does this mean?). If published, this will include your full peer review and any attached files.

Reviewer #2: No

Reviewer #3: No

---

## [Editor Report · Acceptance letter]

1 Jun 2021

PONE-D-21-06261R1 

A cytofluorimetric analysis of a *Saccharomyces cerevisiae* population cultured in a fed-batch bioreactor 

Dear Dr. Sanchez:

I'm pleased to inform you that your manuscript has been deemed suitable for publication in PLOS ONE. Congratulations! Your manuscript is now with our production department. 

Kind regards, 

on behalf of

Dr. Alvaro Galli 

Academic Editor

PLOS ONE